# Act Early and at the Right Location: SARS-CoV-2 T Cell Kinetics and Tissue Localization

**DOI:** 10.3390/ijms231810679

**Published:** 2022-09-14

**Authors:** Antonio Bertoletti, Nina Le Bert, Anthony T. Tan

**Affiliations:** Programme in Emerging Infectious Diseases, Duke-NUS Medical School, 8 College Road, Singapore 169857, Singapore

**Keywords:** adaptive immunity against SARS-CoV-2, tissue-resident T cells

## Abstract

The emergence of new SARS-CoV-2 lineages able to escape antibodies elicited by infection or vaccination based on the Spike protein of the Wuhan isolates has reduced the ability of Spike-specific antibodies to protect previously infected or vaccinated individuals from infection. Therefore, the role played by T cells in the containment of viral replication and spread after infection has taken a more central stage. In this brief review, we will discuss the role played by T cells in the protection from COVID-19, with a particular emphasis on the kinetics of the T cell response and its localization at the site of primary infection.

## 1. Introduction

Since the onset of the COVID-19 pandemic, many critical virological and immunological parameters have changed over time. The emergence of new SARS-CoV-2 lineages able to escape antibodies elicited by previous infection or by vaccination based on the Spike protein of the Wuhan isolates has reduced the ability of Spike-specific antibodies to protect convalescent or vaccinated individuals from (re-)infection [1]. However, despite the increased frequency of breakthrough infections [2], it has become clear that most convalescent and vaccinated individuals are still protected from the development of severe COVID-19 [3]. Therefore, the role that T cells play in the containment of viral replication and spread after infection has taken a more central stage [4,5,6].

At the same time, most of the world population is now no longer immunologically naïve to SARS-CoV-2 and has developed a specific immune response: some individuals through contact with the virus, some through vaccination, and others through a combination of these two events (vaccination and infection, i.e., hybrid immunity). This has led to a mosaic of immune profiles in the population, often stimulated by infection or vaccines based on SARS-CoV-2 lineages that are no longer circulating. In this manuscript, we will discuss the role of T cells in these new scenarios, with a particular emphasis on the kinetics of the T cell response and its localization at the site of primary infection.

## 2. The Role of T Cells during SARS-CoV-2 Infection: Protection or Damage?

During viral infection, CD4 and CD8 T cells execute non-redundant immunological functions that complement the ability of innate immunity to initially contain viral replication and the efficacy of antibodies to prevent infection [7]. CD8 T cells, through the recognition of viral epitopes presented by MHC-class I, detect and eliminate the host cells that harbor a replicating virus. CD4 T cells can also recognize and directly lyse virus-infected cells [8] but they are primarily activated by viral epitopes derived from the processing of viral proteins internalized by professional antigen-presenting cells (mainly dendritic cells or other myeloid lineage cells) and presented by MHC class II molecules. During SARS-CoV-2 infections, SARS-CoV-2-specific CD8 T cells are induced quickly within a few days (4–5 days) after infection [9]. CD4 T cells expand with slightly slower kinetics [10] (reviewed in [6]) and are mainly T follicular helper (Tfh) and Th1 helper CD4 T cells [11,12,13,14]. Tfh cells are primarily required to help B cell proliferation and the production of high-affinity antibodies within the germinal center of secondary lymphoid organs [15]. In contrast, Th1 cells are not involved in B cell maturation and instead support cellular and innate immunity against pathogens [16,17].

While neutralizing antibodies are essential for the prevention of infection, the function of T cells (particularly, CD8 T cells and Th1 cells) is not to prevent infection but to control viral spread and limit viral replication within the infected host by direct lysis of virus-infected cells (CD8 and CD4 T cells) and through the production of antiviral cytokines such as IFN-gamma (CD8 T and Th1 CD4 cells) [18,19]. These functions need to be highly regulated, particularly when the infected cells are in organs that support indispensable physiological tasks, such as the lungs, in order to avoid pathological consequences, since the continuous cell lysis and production of pro-inflammatory cytokines (TNF-alpha and IFN-gamma) can lead to tissue damage [20].

The ability of SARS-CoV-2 infection to trigger a humoral and cellular immune response in infected people was demonstrated within the first few months after the start of the pandemic. Various extended reviews have summarized the main initial findings: neutralizing antibodies directed towards the region of the Spike protein that interacts with the ACE-2 receptor on the surface of cells (called Receptor-Binding Domain—RBD) can prevent infection; CD4 and CD8 T cells specific for structural and non-structural proteins are elicited in infected individuals [6,7,21], and the severity of infection is proportional to the quantity and persistence of SARS-CoV-2 in the airway system [22]. Initial studies that analyzed the relation between the quantities of virus-specific antibodies and T cells and disease severity reported a direct proportional relation between these parameters in convalescent individuals [23,24]. Only subsequent studies that started to analyze adaptive immune parameters longitudinally during the infection showed that a coordinated and early activation of B and T cell responses was associated with viral control with minimal pathologies [10]. Furthermore, the detection of robust and functionally efficient virus-specific T cell responses in SARS-CoV-2-infected but asymptomatic individuals [25,26] started to introduce the concept that virus-specific T cells are an important component of the defense mechanisms of the host against developing COVID-19, something that is already well established for other viral diseases [18,19,27].

Furthermore, mechanistic studies in animal models, such as the demonstration of a substantial reduction in the ability of antibodies to control the viral infection in CD8 T cell-depleted SARS-CoV-2 convalescent [28] or vaccinated [29] macaques or the ability of CD4 T cells to reduce lung pathologies in a mouse model of SARS-CoV-2 [30] clearly showed the importance of SARS-CoV-2 T cells in protecting from the pathological consequences of infection.

One of the best examples that demonstrated the importance of a rapid activation of both humoral and cellular immunity in the control of the natural SARS-CoV-2 infection was derived from a longitudinal study of acutely infected patients who developed mild and severe COVID-19 [31]. By studying a limited number of patients, Tan et al. showed that a rapid expansion of IFN-gamma-producing T cells specific for different structural and non-structural SARS-CoV-2 proteins, including Spike (S), membrane (M), nucleocapsid protein (NP) and ORF7/8, coupled with antibody production was detectable only in patients who rapidly controlled SARS-CoV-2 replication without severe pathological consequences. In contrast, patients with prolonged infection and severe COVID-19 mounted antibody responses, even at levels higher than in patients with mild COVID-19, but had undetectable circulating SARS-CoV-2 T cells [31]. This initial study, which supported the idea that T cells offer an important contribution to the protection against COVID-19, i.e., disease development, was confirmed subsequently in larger longitudinal studies of both asymptomatic or pauci-symptomatic patients [32] and mild and severe COVID-19 patients [33].

Other evidence that supports an important role of the early activation of virus-specific T cells in the rapid reduction of viral replication and the attenuation of disease severity derived from studies that detected cross-reactive SARS-CoV-2-specific T cells, likely primed by seasonal Coronaviruses, in individuals with an abortive infection or mild COVID-19. For example, increased frequencies of T cells specific for polymerase (NSP12) [34] or other SARS-CoV-2 proteins [35] were detected in individuals who successfully aborted infection (highly exposed individuals who remained PCR-negative for SARS-CoV-2 and seronegative). Importantly, in the work of Swadling et al., the activation of type-I IFN genes, classically associated with SARS-CoV-2 infection and initial virus replication, was detected in the individuals who then demonstrated an expansion of NSP-12-specific T cells, suggesting that the presence of such SARS-CoV-2-specific T cells was able to immediately abort virus replication, since these individuals remained SARS-CoV-2-PCR negative and did not produce any antibody response. Similarly, studies showed that less severe COVID-19 was linked with a recent exposure to seasonal Coronaviruses [36] and demonstrated that the presence of cross-reactive T cells facilitated the expansion of SARS-CoV-2-specific CD8 and CD4 T cell responses during infection [37]. However, these data are still controversial, and the impact of cross-reactive T cells on altering SARS-CoV-2 infection has not been determined in all studies [38,39].

Also in vaccinated individuals, the early presence of Spike-specific T cells was correlated with the reduction of COVID-19 symptoms. By tracking the humoral and cellular immune responses after the first dose of mRNA vaccines, Kalimuddin et al. demonstrated that Spike-specific T cells and non-neutralizing antibodies were detected as early as day 10 and preceded the detection of neutralizing antibodies, which became detectable only around week 3 [40].

Thus, an ample body of experimental evidence, both in animal models and in naïve or vaccinated and/or infected individuals, supports the role of SARS-CoV-2-specific T cells in the protection from disease development. The importance of CD4 and CD8 T cells in controlling viral pathogenesis is of particular interest in relation to the emergence of new SARS-CoV-2 variants that are mainly mutated in the Spike protein. The new Omicron variants are, for example, able to escape the neutralization effect of most of the Spike-specific antibodies induced by infection or vaccination [1]. However, SARS-CoV-2 infection induces virus-specific CD4 and CD8 T cells specific for different epitopes on both structural and non-structural proteins [24,31,41,42] that tolerate most, even though not all, the AA mutations present in newly emergent Omicron lineages [43,44,45]. Similarly, though mRNA- and adenovirus-based vaccines induce only Spike-specific CD4 and CD8 T cells [46,47,48,49,50], the vaccine-triggered memory Spike-specific T cell response is multi-epitope-specific and can also tolerate most of the Omicron variations [44,46,51,52]. Similar data have also recently been demonstrated for T cells induced by vaccines based on inactivated viruses that stimulate not only Spike-, but also Membrane- and NP-specific T cells. Even though the inactivated viral vaccines trigger a preferential CD4 T cell response, they appear to tolerate extremely well Omicron mutations present in Spike but also in NP and Membrane [53]. This can explain why the high frequency of breakthrough infections with Omicron in vaccinated individuals is not associated with a loss of efficacy in the protection from severe disease in both mRNA-vaccinated [2,54] and inactive virus-vaccinated individuals [55].

## 3. T Cell Response and Pathology

Despite the data linking the early activation and presence of virus-specific cellular immunity with protection from severe disease, we must also consider the possibility that a persistent and heightened activation of T cells might also be associated with pathogenic damage. While severe COVID-19 is associated with lymphopenia [56], persistent inflammatory events [57,58,59,60] with detection of T cell activation are present not only in the circulation but also in pathological tissues. Despite the presence of lymphopenia, activated CD8 T cells have been found, for example, in the lung and also in the brain of patients with severe COVID-19 [61], but it is important to highlight that T cells are not the only population that is present in high number and with an activated phenotype in the pathological tissues. For example, increased plasmablast frequencies [59] and large numbers of activated myeloid cells showing type-I IFN gene activation were found to characterize the lung pathological infiltrate in severe COVID-19 patients [62,63]. Interestingly, recent work has shown that inflammatory events mediated by myeloid lineage cells [64] can be sustained by an antibody-mediated uptake of the virus [65], and overall, a heightened activation of myeloid cells is a constant signature of severe COVID-19. When lung and blood samples of patients with severe COVID-19 were analyzed in parallel, higher T cell frequencies in the lung positively correlated with survival, while conversely, higher quantity of myeloid cells in the lung correlated with mortality [62]. Furthermore, it is important to point out that the antigen specificity of the T cells detected in tissues during severe COVID-19 has never been demonstrated, and their expression of activation or exhaustion markers might be caused by bystander immunological events and not by TCR-mediated antigen-specific recognition [66,67]. Another important point is that the prolonged and persistent respiratory symptoms that follow COVID-19 (the so-called long COVID) are associated with an increased presence of composite immune infiltrates in the lung in which T cells expressing markers of activation and exhaustion can also be found [68].

However, more recent work has also highlighted the importance not just of the quantity but also of a regulated and coordinated sequential modification of CD4 T cell function in the processes of tissue inflammation and control of viral replication. A conversion of CD4 T cells from a mainly-Th1 phenotype towards an IFN-gamma and IL-10 functional profile was observed in the mechanism of viral control without overt and persistent pathology [69,70]. Mouse models of viral lung infection have, for example, shown the importance of IL-10-producing T cells in shaping the ability of T cells to control the virus without triggering an overt lung pathology [71,72]. Of note, asymptomatic SARS-CoV-2-infected patients demonstrated such a T cell cytokine profile [26]. On the other hand, the inability of Th1-polarized CD4 T cells to switch towards an IL-10-producing phenotype has been shown to cause severe tissue damage in different infection models [73], and a prolonged Th1 cytokine profile has been demonstrated in the blood of patients with severe COVID-19 [60] and in the bronchoalveolar lavage during ongoing lung inflammation [69].

Importantly, the resolution of the pathological processes of lung inflammation was associated with the progressive shutdown of Th1 cytokine production and induction of IL-10 [69], a regulatory process apparently orchestrated by the increased presence of vitamin D and activated by the complement produced by respiratory epithelial cells [74]. This autoregulatory process appears deficient in patients with severe COVID-19, and interestingly, the severity of COVID-19 has been associated with vitamin D insufficiency in an epidemiological study [75].

It is therefore possible that mechanisms of functional dysregulation in T cells might play a role in the exacerbated inflammatory events that characterize severe COVID-19 and even in some aspects of the prolonged pathology observed in some COVID-19 convalescents [68]. Certainly, more studies are needed to analyze the functionality of SARS-CoV-2-specific T cells during these pathological events, studies that are still lacking also because, as we have summarized, these pathological processes are usually associated with a quantitative and/or functional deficiency of SARS-CoV-2-specific T cells.

## 4. Act at the Right Place: SARS-CoV-2-Specific T Cells in the Upper and Lower Airways

Most of the current analysis of the cellular and humoral immune response has been limited to the circulatory compartment. However, blood is not the primary site of SARS-CoV-2 infection, and except in cases of severe COVID-19, SARS-CoV-2 is rarely detectable in the circulation [76].

The role and the importance of virus-specific T cells resident in the specific tissues targeted by different viruses has been already reported: for example, in infections of the skin and genital mucosa, tissue-resident virus-specific CD8 T cells act as a first layer of protection and trigger components of the innate and adaptive immunity that achieve a rapid “near-sterilizing” immunity [77]. Similarly, in infections with influenza or respiratory syncytial viruses, both the presence and the adoptive transfer of tissue-resident CD8 T cell in the nasal cavity control viral spread and disease severity [78,79]. Lung-resident virus-specific T cells producing IL-10 have also been associated with influenza virus clearance with limited lung pathology [71].

It makes perfectly good sense that we should therefore start to analyze in more detail the characteristics of the immune responses present in the upper and lower airways. Indeed, the initial anatomical site of SARS-CoV-2 infection and replication is the nasal cavity [57,58,59]. Due to their elevated ACE-2 receptor expression, nasal ciliated cells are primarily infected and sustain the majority of initial virus production in vivo [80]. SARS-CoV-2 can then colonize the epithelial cells of the lower respiratory tract [81], even though it has been also detected in other extra-pulmonary sites, e.g., liver, kidney, heart, brain, gut [82,83,84].

The importance of SARS-CoV-2-specific T cells in the airway system has been first demonstrated in animal models of Coronavirus infection. In mice infected with coronaviruses (MERS-CoV and SARS-CoV), the activation of coronavirus-specific memory CD4 T cells resident in the upper airways was necessary for viral clearance. In this model, the secretion of IFN-gamma in the lung tissue by activated CD4 T cells enhanced the recruitment of SARS-CoV-2-specific CD8 T cells [72]. More recent works in mice [85,86,87] or hamsters [88] vaccinated with novel vaccine preparations eliciting a SARS-CoV-2-specific immune response in the upper airways have provided additional data. In some of these models, the induction of mucosal specific antibodies clearly contributed to the observed protection [86,87,88], but in others, the role played by vaccine-induced mucosal T cell responses in the protection was evident [85].

Studies related to SARS-CoV-2-specific T cells in the airways, the overall primary site of infection, in humans are still limited [89,90,91,92,93,94,95]. During acute SARS-CoV-2 infection, dynamic modifications of immune cell populations in different airway tissues (nasal, oropharyngeal cavity and lung tissues) have been described, with a preferential recruitment of granulocytes, inflammatory monocytes, macrophages and NK cells [94]. However, T cell enrichment was also detected, likely triggered by elevated levels of chemokines secreted into the tissues which were proportional to the levels of SARS-CoV-2 replication [96]. After resolution of the SARS-CoV-2 infection, tissue-resident SARS-CoV-2-specific memory T cells have been demonstrated. They can be found in different tissues (bone marrow, spleen, lung, and lymph nodes) [93] and persist for at least 6 months after infection. SARS-CoV-2 infection has also been shown to induce peripheral virus-specific T cells with phenotypic characteristics of tissue homing [97].

Moreover, tissue-resident Spike-specific CD8 T cells were also observed in the nasal cavity after resolution of an acute SARS-CoV-2 infection [94], demonstrating that the infection does induce a tissue-localized antigen-specific T cell response.

It is however important to remember that tissue-resident SARS-CoV-2-specific T cells have been also described in the oropharyngeal tonsils and in BAL specimens of healthy individuals who were never in contact with SARS-CoV-2 [90,91]. These SARS-CoV-2-specific T cells were detected in more than half of the tested individuals, they expressed classical tissue-resident phenotype markers (CD103+ and CD69+) and produced Th1 cytokines with a preferential secretion of TNF-alpha and a low secretion of IFN-gamma [90,91]. Importantly, these cross-reactive tonsil-resident SARS-CoV-2 T cells displayed, in adults but, apparently, not in children, a lower capacity to secrete Th1 cytokines than CD8 T cells specific for other viruses such as EBV, CMV and seasonal coronavirus (HCoV-OC43) [90,91]. Since these T cells specific for different epitopes located in structural and non-structural proteins were likely induced by infections with other seasonal coronaviruses, this partial functionality might be caused by their lower affinity for the SARS-CoV-2 peptides used for stimulation, a reported characteristic of some cross-reactive CD4 T cells [98]. Whether these tissue-resident cross-reactive T cells have a protective effect is therefore still debatable, and more independent studies are needed to understand whether such findings can be confirmed.

Nevertheless, these studies show that cross-reactive SARS-CoV-2 T cells are detectable also in tissues and not only in the circulation and imply that infection with coronaviruses might be necessary for the presence of tissue-resident SARS-CoV-2-specific T cells. A lingering question is whether parenteral vaccines that have been used world-wide can induce or boost a Spike-specific T cell response in airway tissue. Parenteral vaccinations clearly induce good T and B cell response in the blood [46,99,100,101,102,103], but detailed studies related to their preferential localization showed an extremely localized B and T cell response in the lymph node draining the site of infection [104,105].

One study [89] reported a very high frequency (up to 20% of total T cells) of vaccine-induced Spike-specific T cells in the nasal cavity of vaccinated individuals, 2 and 4 months after vaccination. This high induction of Spike-specific T cells in the nasal cavity by parenteral vaccination has not, however, been confirmed by more recent studies. Spike-specific T cells were, for example, detected at high frequencies in BAL samples of vaccinated individuals following a breakthrough infection, but not in vaccinated-only individuals [92].

We have also recently characterized the SARS-CoV-2 specificity of T cells purified from the nasal cavity of individuals who were either only vaccinated or also experienced a breakthrough Omicron infection. We were able to detect tissue-resident (CD103+ CD69+) CD4+ and CD8+ T cells specific for Spike and other SARS-CoV-2 proteins only in vaccinated individuals who suffered a breakthrough infection. No SARS-CoV-2-specific T cells (specific also for Spike) were found in healthy COVID-19 vaccinees [95]. It seems therefore that infection, and not parenteral vaccination, is necessary to trigger a robust tissue-localized SARS-CoV-2-specific T cell response, a notion that is also in line with the different phenotypes of circulating Spike-specific T cells observed in vaccinated naïve or convalescent individuals [67]. Only convalescent vaccinated individuals showed Spike-specific T cells with a tissue-homing phenotype [67].

That T cells specific for different SARS-CoV-2 antigens were only detected in the nasal cavity of vaccinated individuals with breakthrough infection [95] is in apparent contrast with the demonstration of SARS-CoV-2 cross-reactive T cells in the tonsils and BAL of healthy individuals [90,91]. However, virus-specific T cells in the lung and oropharyngeal cavity are differentially regulated compared to the ones present in the nasal cavity [78,106]. Animal models of viral respiratory infections demonstrated that nose-associated lymphoid tissues are not the site of induction of virus-specific T cells, but they recruit and support the persistence of tissue-resident T cells specific for respiratory viruses, primed initially in the more organized lymphoid organs present in the oral cavity (oropharyngeal and palatine lymph nodes) [78,106]. It is therefore possible that the detection of a wide repertoire of T cells specific for different SARS-CoV-2 proteins (NP, Spike, M and NSP-12) observed by us in the nasal cavity of vaccinated individuals who experienced a breakthrough infection was the result of a specific recruitment, triggered by the localized infection, of T cells primed in different anatomical locations [95]. These nasal-resident SARS-CoV-2-specific CD4 and CD8 T cells produce a high quantity of IFN-gamma, differently from what was observed in the tonsils of healthy individuals [90,91]. Importantly, the detection of T cells able to recognize different epitopes located in multiple non-Spike proteins also in tissues and not only in the circulatory compartment [107,108,109] demonstrated that vaccination does not suppress the induction of a broader SARS-CoV-2-specific T cell repertoire after infection.

This is important, since some recent data have suggested a reduced ability of Omicron infection to induce a robust T and B cell response in patients who were previously infected or vaccinated [110]. The reduced Omicron immunogenicity was, however, only measured against the S1 region of Spike and not against the whole SARS-CoV-2 proteome, and natural infection in vaccinated-only individuals primed a broad SARS-CoV-2-specific T cell response both in the circulation [109] and in the nasal cavity [95]. We think that multispecificity and tissue localization of SARS-CoV-2-specific T cell constitute an important feature of individuals with hybrid immunity, i.e., subjects who were vaccinated and infected. Clinical data with a careful analysis of the viral load in the early phase of infection in individuals who suffered an Omicron breakthrough infection showed that a more rapid control of virus replication was evident in vaccinated individuals with hybrid immunity (vaccination and infection) compared to individuals vaccinated with three doses of vaccines [111]. Very recent epidemiological data have also shown that the protection induced by infection is also superior to that induced by mRNA vaccination alone [3].

The presence of nasal T cells is not, however, the only characteristic of individuals with hybrid immunity. Infected and vaccinated individuals can produce a broader repertoire of antibodies [108,112] and Spike-B cells [109,113], and their T cells not only recognize multiple proteins [107,108,109] but also are characterized by a functional profile of IFN-gamma and low IL-10 production, that characterizes asymptomatic SARS-CoV-2-infected individuals [26] and, in mice, is associated with protection from infection without overt lung pathology [71,72].

## 5. Final Remarks

Changes of the viral genome occurred during the COVID-19 pandemic and resulted in SARS-CoV-2 lineages that largely escape the humoral immunity induced by vaccines based on the Spike protein of the ancestor Wuhan isolates [2]. However, vaccines (and previous infections) are still able to protect most individuals from severe pathologies. Like others [5], we think that a comprehensive analysis of the antiviral immunity triggered by vaccination, infection or both should not be exclusively based on antibody measurements, but also evaluate the virus-specific cellular immunity at different anatomical locations, since many data show its association with protection from disease development (Figure 1).

Herein, we reviewed aspects of the SARS-CoV-2-specific T cell immunity that are related to protection from disease development. However, more data on T cell responses in large clinical trials of vaccine protection in different populations are needed to better define the role of T cells in SARS-CoV-2 infection. This could be facilitated by new methods that can provide a rapid estimation of the quantity, diversity and function of SARS-CoV-2-specific T cells in samples [114,115,116,117] and by animal models that recapitulate the human SARS-CoV-2 infection more precisely [21,118]. As we have recently argued [119], the evaluation of the role of T cells in SARS-CoV-2 infection will need to consider also the strategies that SARS-CoV-2 might possess to avoid T cell recognition, not only sequence mutations but also active alterations of antigen processing and presentation. In this regard, evidence is starting to emerge from different publications [120,121] and will need to be defined in major detail to better understand the overall impact of T cells on disease protection and whether new vaccination boosts are necessary.

## Figures and Tables

**Figure 1 ijms-23-10679-f001:**
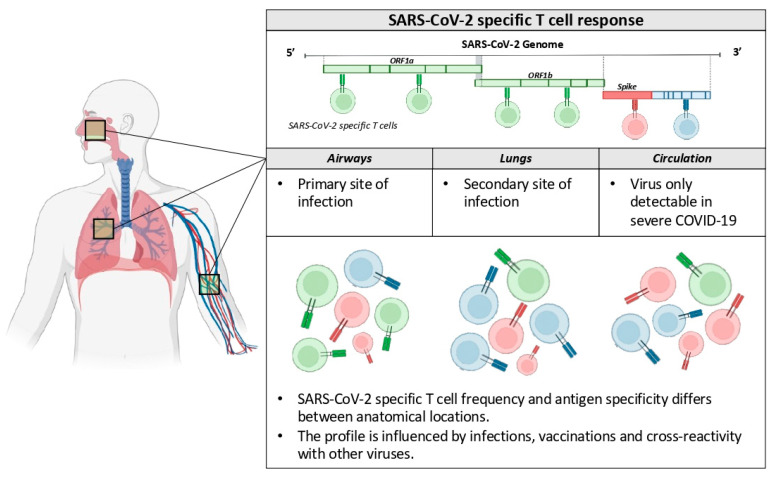
SARS-CoV-2-specific cellular immunity at different anatomical locations.

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
