# Peer review of "Act Early and at the Right Location: SARS-CoV-2 T Cell Kinetics and Tissue Localization"

_ijms, 2022, doi:10.3390/ijms231810679_

Round 1

Reviewer 1 Report

The fundamentally concepts about T cells activation a correctly formulated. A lot of results reported in time, a short protective period of antibodies. It is a general agree that cell mediated IR is much more protective against SARS CoV-2 infection.

Lingering the protective efficiency of cellular immunity postvaccination or postinfection, is difficult to conclude. A lot of people are reinfected, while other vaccinated only are protected. IR against SARS CoV-2 is dependent on very many individual factors and can't formulate valuable conclusions.You propose a differential analysis of T cell memory populations at the levels of respiratory tract, but the T cells recirculate... !

Author Response

We are sorry that the reviewer 1 found our manuscript of minimal interest and not scientifically sound ( judging from the number of stars..). We have also difficuties to understand the comment about the ability of T cells to recirculate ( You propose a differential analysis of T cell memory populations at the levels of respiratory tract, but the T cells recirculate... !). As we tried to explain in the review and in the paper just published in JEM (Lim, J. M. E. et al. SARS-CoV-2 breakthrough infection in vaccinees induces virus-specific nasal-resident CD8+ and CD4+ T cells of broad specificity. J Exp Med 219, e20220780 (2022), the presence of Tissue resident T cells in the nasal cavity can provide an important first layer of protection against viruses and immediately blocking viral replication and spread. The importance of such first layer of T cells located at the site of primary infection has been shown to be important in many animal models. ( See for example Schenkel, J.M., K.A. Fraser, L.K. Beura, K.E. Pauken, V. Vezys, and D. Masopust. 2014. T cell memory. Resident memory CD8 T cells trigger protective innate and adaptive immune responses.Science . 346:98–101. doi:10.1126/science.1254536), despite the fact that in these models T cells are, as you wrote , recirculating. In any case we added the new papers published by different groups about SARS-CoV-2-specific T cells in the respiratory tract to further support our claim of the importance of such cells for SARS-CoV-2 control. 

Reviewer 2 Report

In this brief review “ Act early and at the right location: SARS-CoV-2 T cells kinetics  and tissue localization” the authors highlight the T cell responses after infection and vaccination and localization.

This brief review in general is well written but it Will Benefit from some minor changes:

-In this paragraph:

 “CD8 T cells, through 37 recognition of viral epitopes presented by MHC-class I, detect and eliminate the host 38 cells that harbor replicating virus. CD4 T cells, on the other hand, do not directly recognize infected cells but are instead activated by viral epitopes derived from the pro-40 cessing of viral proteins internalized by professional antigen-presenting cells (mainly 41 dendritic cells or other myeloid-lineage cells) and presented by the MHC-class II molecules”

It lacks references but also, CD4+ T cells are also cytotoxic: Juno JA, van Bockel D, Kent SJ, Kelleher AD, Zaunders JJ, Munier CM. Cytotoxic CD4 T Cells-Friend or Foe during Viral Infection? Front Immunol. 2017 Jan 23;8:19. doi: 10.3389/fimmu.2017.00019.

-Line 50-58 need references

-In this paragraph: The ability of SARS-CoV-2 infection to trigger a humoral and cellular immunity in the infected hosts was immediately demonstrated few months after the start of the pandemic.

I think would be important to highlight the role of memory T cells to control re infections, since some of the new variants can evade the humoral response, and also memory T cells produce a more rapid response than naïve T cells and react to many SARS-COV_2 epitopes so the evasion from immune response is less likely doi: 10.3389/fcell.2021.620730

-Line 76: established in other viral diseases[23].

Please add more references

-Authors may want to say something about lymphopenia in introduction, Total lymphocytes include not only T cells but lymphopenia ia marker of disease severity in covid19 and lymphopenia hampers a robust T cell response against the virus

Author Response

We thank the reviewer for the general positive comments in relation to the content of our manuscript. We modified the manuscript to include your suggestions. 

Point 1) It lacks references but also, CD4+ T cells are also cytotoxic:

We agree about this comment and we modified the text to point out that CD4 T cells can also directly recognised viral infected cells and added new references : 

McKinstry, K.K.; Strutt, T.M.; Kuang, Y.; Brown, D.M.; Sell, S.; Dutton, R.W.; Swain, S.L. Memory CD4+ T Cells Protect against Influenza through Multiple Synergizing Mechanisms. Journal of Clinical Investigation 2012, 122, 2847–2856, doi:10.1172/jci63689.

17. Strutt, T.M.; McKinstry, K.K.; Dibble, J.P.; Winchell, C.; Kuang, Y.; Curtis, J.D.; Huston, G.; Dutton, R.W.; Swain, S.L. Memory CD4+ T Cells Induce Innate Responses Independently of Pathogen. Nature Medicine2010, 16, 558–564, doi:10.1038/nm.2142.

Point 2) -Line 50-58 need references

We added new references as requested:  

18. Chisari, F.V. Cytotoxic T Cells and Viral Hepatitis. J Clin Invest 1997, 99, 1472–1477, doi:10.1172/jci119308.

19. Schmidt, M.E.; Varga, S.M. The CD8 T Cell Response to Respiratory Virus Infections. Front Immunol 2018,9, 678, doi:10.3389/fimmu.2018.00678.

20. Bertoletti, A.; Maini, M.K. Protection or Damage: A Dual Role for the Virus-Specific Cytotoxic T Lymphocyte Response in Hepatitis B and C Infection? Curr Opin Immunol 2000, 12, 403–408, doi:10.1016/s0952-7915(00)00108-4.

Point 3) I think would be important to highlight the role of memory T cells to control re infections, since some of the new variants can evade the humoral response, and also memory T cells produce a more rapid response than naïve T cells and react to many SARS-COV_2 epitopes so the evasion from immune response is less 

We discussed in the paragraph at page...the impotence to establish a multi proteins and multi-epitopes specific memory T cell response against SARS-C0V-2 variants. We highlighted the establishment of tissue-memory T cell response after infection  ( particulalry in the airway) in order to rapidly control SARS-CoV-2 infection. 

Point 4) Line 76: established in other viral diseases[23].

Please add more references

We added more references: 

18. Chisari, F.V. Cytotoxic T Cells and Viral Hepatitis. J Clin Invest 1997, 99, 1472–1477, doi:10.1172/jci119308.

19. Schmidt, M.E.; Varga, S.M. The CD8 T Cell Response to Respiratory Virus Infections. Front Immunol 2018,9, 678, doi:10.3389/fimmu.2018.00678.

POINT 5 -Authors may want to say something about lymphopenia in introduction, Total lymphocytes include not only T cells but lymphopenia ia marker of disease severity in covid19 and lymphopenia hampers a robust T cell response against the virus

We introduced the concept of lymphopenia in the paragraph of 

"T cell response and pathology" where we highlighted that severe COVID-19 is associated with lymphopenia ( ref 51)